# *hTERT* DNA Methylation Analysis Identifies a Biomarker for Retinoic Acid-Induced *hTERT* Repression in Breast Cancer Cell Lines

**DOI:** 10.3390/biomedicines10030695

**Published:** 2022-03-17

**Authors:** Eric Nguyen, Andréa Richerolle, Júlia Sánchez-Bellver, Jacqueline Varennes, Evelyne Ségal-Bendirdjian

**Affiliations:** 1Université Paris Cité, INSERM, CNRS, T3S “Environmental Toxicity, Therapeutic Targets, Cellular Signaling and Biomarkers”, F-75006 Paris, France; enguyen1@club-internet.fr (E.N.); andrea.richerolle@gmail.com (A.R.); jacqueline.varennes@inserm.fr (J.V.); 2Ecole Pratique des Hautes Etudes, F-75014 Paris, France; 3Universitat de Barcelona, 08028 Barcelona, Spain; juliasbellver@gmail.com

**Keywords:** telomerase, *TERT*, DNA methylation, epigenetic, breast cancer, ATRA

## Abstract

Telomerase reactivation is responsible for telomere preservation in about 90% of cancers, providing cancer cells an indefinite proliferating potential. Telomerase consists of at least two main subunits: a catalytic reverse transcriptase protein (*hTERT*) and an RNA template subunit. Strategies to inhibit *hTERT* expression seem promising for cancer treatment. Previous works showed that all-*trans* retinoic acid (ATRA) induces *hTERT* repression in acute promyelocytic leukemia cells, resulting in their death. Here, we investigated the effects of ATRA in a subset of breast cancer cell lines. The mutational status of *hTERT* promoter and the methylation patterns at a single CpG resolution were assessed. We observed an inverse relationship between *hTERT* expression after ATRA treatment and the methylation level of a specific CpG at chr5: 1,300,438 in a region of *hTERT* gene at −5 kb of the transcription initiation site. This observation highlighted the significance of this region, whose methylation profile could represent a promising biomarker to predict the sensitivity to ATRA-induced *hTERT* repression in specific breast cancer subtypes. As *hTERT* repression promotes drug-induced cell death, checking the methylation status of this unique region and the specific CpG included can help in decision-making to include ATRA in combination therapy and contributes to a better clinical outcome.

## 1. Introduction

Telomeres are the terminal ends of linear chromosomes. In most eukaryotes, they are composed of a DNA repeat sequence (TTAGGG) capped by telomere-specific binding proteins [1]. Telomeres shorten progressively with each cell division, causing cell senescence and eventually cell death [2]. Therefore, maintaining telomere length is critical for cancer cells to survive and keep proliferating [3].

Most cancers activate telomerase to maintain telomere length [4], thereby avoiding senescence and apoptosis, and leading to indefinite replicative and proliferative capacity [5]. Human telomerase consists of at least two main components, an enzymatic subunit telomerase reverse transcriptase (*hTERT*) and an RNA subunit (hTR), which is used as a template for the elongation of telomeric DNA [6,7].

Not only does telomerase have a role in preventing telomere shortening and cell senescence in highly proliferative cells, but also studies including ours show that telomerase exhibits additional roles independent of its activity on telomere length maintenance. These non-conventional roles include anti-apoptotic functions, activation of proliferative signaling pathways, and resistance to anticancer therapies [8,9]. Therefore, telomerase represents an attractive target for selective cancer therapeutics [10].

The limiting component for the control of telomerase activity is the *hTERT* catalytic subunit [11]. It is regulated at many levels including transcription, which is the primary determinant of telomerase activity. Several mechanisms are associated with *hTERT* reactivation including genomic rearrangements [12], gene copy number variations [13], and *hTERT* gene promoter point mutations [14,15]. Indeed, two mutually exclusive mutation hotspots are located in the *TERT* promoter: chromosome 5p15.33:1,295,228 C > T and 1,295,250 C > T, usually referred to as C228T and C250T, respectively (G > A on the opposite strand). These mutations create additional binding sites for the E-twenty-six (ETS) transcription factor family [16,17]. They modulate transcriptional regulation without altering the encoded protein. In addition to genetic mechanisms, the transcription of *hTERT* is also regulated by epigenetic mechanisms including *TERT* promoter methylation and histone post-translational modifications [18]. 

All-trans-retinoic acid (ATRA) is a non-conventional and promising therapeutic agent, known for its cell differentiation-inducing properties. It is the current therapy for acute promyelocytic leukemia (APL) [19,20]. In APL patients, ATRA induces the differentiation of leukemic cells. In APL cells, long-term treatment with ATRA downregulates *hTERT*, leading to cell death through a telomere/telomerase-dependent mechanism even in cells resistant to differentiation induced by this drug [21,22,23,24]. Concerning this later cellular model, we identified two distinct functional domains in the *hTERT* promoter: a proximal domain including the core promoter of *hTERT* (−200 bp to −150 bp), which encodes response elements or binding sites essentially for activators of *hTERT* expression, and a more distal domain, upstream of the core promoter (−600 bp to −200 bp), where *hTERT* repressor signaling likely converges [25]. This latter domain included a region that was later referred to as the *TERT* hypermethylated oncological region (THOR) [26], which can play a role as a potential diagnostic and prognostic marker in different cancer types [27,28,29]. Moreover, further studies performed in APL cells highlighted the significance in activating *hTERT* of an unexplored domain outside the minimal *hTERT* promoter, localized around 5 kb upstream from the transcription start site [30].

Pharmaco-clinical works performed in different cellular contexts have shown that the pharmacological modulation of *hTERT* expression by ATRA can be used therapeutically to sensitize cells to anti-cancer therapies and/or to overcome drug resistance responsible for the failure of therapies [8,31]. Therefore, the above observations can provide the basis for further works to extend these findings and translate them into promising new approaches for the treatment of a broad range of cancers including breast cancers. 

Breast cancer is a heterogeneous disease encompassing a group of genetically and epigenetically distinct diseases exhibiting diverse clinical features. Tissue-based biomarkers have been central to tumor subtyping, prognostic, and choice of therapies. The clinical classification of breast cancer currently relies on histological grading, hormone receptor status, and molecular classification. Breast cancer is classified into distinct molecular subtypes according to the presence/absence of the estrogen receptor (ER), the progesterone-receptor (PR), the human epidermal growth factor receptor 2 (HER2), and specific gene expression profiles. Triple-negative breast cancers (TNBC, 15–20% of breast cancers) lack ER and PR and do not overexpress HER2. ER ^+^/PR^+^ breast cancers (60% of breast cancers) can be treated with endocrine therapies such as tamoxifen and/or aromatase inhibitors to block the production of estrogens [32]. Despite the efficacy of tamoxifen treatment, a third of ER^+^ treated breast cancers relapse. HER2 breast tumors represent a distinctive group as it is characterized by HER2 overexpression/amplification, which is a major driver of their pathogenesis. It results in the constitutive activation of HER2 signaling pathways. Tumors of this specific group are oncogene addicted due to their dependency on HER2 functions [33,34,35]. This group of breast cancers (10–15% of breast cancers) can benefit from anti-HER2 antibodies such as trastuzumab (Herceptin). While HER2-targeted therapy revolutionized outcomes in HER2^+^ breast cancer, many patients are either initially resistant or acquire resistance to this therapy, leading to disease progression. TNBC have the poorest prognosis. They have no targeted therapies (excepted BRCA1 mutated TNBC that can be treated with PARP inhibitors) and are currently treated with conventional chemotherapy [36]. Telomerase activity has been detected in around 90–95% of invasive breast cancers, whereas no activity has been found in non-malignant breast tissues [37,38,39]. A high expression of *hTERT* was strongly associated with lower overall survival in breast cancer patients, suggesting its potential use as a diagnostic/prognostic marker [38,40]. 

Even though all the mechanisms of action of ATRA are incompletely understood, the available pre-clinical data have raised interest in this drug for the treatment of breast cancer [41,42,43,44]. However, in the few clinical trials conducted, the therapeutic benefit remained limited. This low efficacy can be attributed to the pathogenic complexity of these cancers, and to the fact that the trials were performed on patients not selected on the basis of the subtype and grade of the cancers. Moreover, in these studies, only the cytodifferentiating effects of ATRA were considered. However, it has now been shown that the restoration of differentiation alone does not explain the success of this therapy [45]. 

Based on the previous work performed by our group on APL cells, we propose in this study to take into account the anti-telomerase properties of ATRA and to identify factors that could predict the cellular response to this drug. Therefore, we investigated the consequences of ATRA treatment on *hTERT* expression in a subset of cultured breast cancer cells belonging to different clinical subtypes. In addition, we presented detailed profiles of *hTERT* methylation at a CpG-specific resolution obtained by the “gold standard” bisulfite DNA sequencing method and explored the associations between *hTERT* methylation patterns and sensitivity to ATRA-induced repression of *hTERT*. We found a statistically high significant negative correlation between the methylation levels of a region of the *hTERT* gene located 5 kb upstream of the TSS and the effect of ATRA on *hTERT* mRNA expression. These data demonstrate that differential methylation of specific CpG sites may be useful biomarkers to predict ATRA-induced *hTERT* repression.

## 2. Materials and Methods

### 2.1. Cell Culture and ATRA Treatment

A panel of ten different breast cancer (BC) cell lines representing major breast cancer subtypes were enrolled in the study including ER-positive (ER^+^), HER2 positive (HER2^+^), and triple-negative (TN) subtypes (Table 1). They were cultured for fewer than six months from the time of resuscitation. SKBR3 and BT474 cells were obtained from the American Type Culture Collection (ATCC, LGC Standards, Molsheim, France) and MDA-MB-361 cells from Sigma Aldrich Chimie S.a.r.l. (Saint-Quentin-Fallavier, France). They were authenticated by the suppliers. All the other cell lines were authenticated by short tandem repeat analysis (STR) using the Promega Powerplex 21 PCR Kit (Eurofins, MWG Biotech, Ebersberg, Germany). MCF7, MDA-MB-453, and MDA-MB-231 cells were cultured in DMEM 4.5 g/L glucose supplemented with 10% fetal calf serum, 1% Zellshield (Minerva-Biolabs, Biovalley, Nanterre, France ), 1% glutamine (200 mM); T47D and BT474 cells were cultured in RPMI supplemented with 10% fetal calf serum, 1% Zellshield, 1% glutamine (200 mM); ZR75.1, SKBR3, and MDA-MB361 cells were cultured in DMEM:Ham’s F12 (1/1) supplemented with 10% fetal calf serum, 1% Zellshield, 1% glutamine (200 mM) 1% bicarbonate (200 mM); SUM159PT cells were cultured in DMEM:Ham’s F12 (1/1) supplemented with 10% fetal calf serum, 1% Zellshield, 1% glutamine (200 mM); SUM185PE were cultured in DMEM:Ham’s F12 (1/1) supplemented with 10% fetal calf serum, 1% Zellshield, 1% glutamine (200 mM), 1% bicarbonate (200 mM), and 5 µg/mL insulin. All culture mediums, glutamine, insulin, and bicarbonate were from Fisher Scientific S.A.S., Illkirch, France. Fetal calf serum was from PAA Laboratories (Pasching, Austria). All cells were incubated at 37 °C and 5% CO_2_ and harvested using 0.25% trypsin-EDTA (Fisher Scientific S.A.S., Illkirch, France). When treated, cells were incubated with 1 µM ATRA (Sigma Aldrich Chimie S.a.r.l., Saint-Quentin-Fallavier, France) for seven days. Half of the medium was renewed three days after the beginning of the treatment.

### 2.2. RNA Extraction and Quantitative Reverse Transcriptase Polymerase Chain Reaction (qRT-PCR)

Total RNA was extracted from cell lines using Trizol (Ambion), according to the manufacturer’s instructions, and subjected to reverse transcriptase reaction with random hexamer primers using the Transcriptor First Strand cDNA Synthesis Kit (#04 897 030 001, version 6, Roche Diagnostics, Meylan, France) as described in the manufacturer’s instructions. Subsequently, the cDNAs were submitted to quantitative real-time PCR using the LightCycler technology and the Light Cycler FastStart DNA MasterPLUS SYBR Green Kit (#03 753 186 001, Roche Diagnostics). The endogenous housekeeping genes glyceraldehyde-3-phosphate dehydrogenase (*GAPDH*) and porphobilinogen deaminase (*PBGD*) were used as normalization controls. Primer sequences are shown in Appendix A.

### 2.3. DNA Extraction and Sanger Sequencing

The presence of *hTERT* promoter/enhancer mutations was evaluated by conventional Sanger sequencing. Genomic DNA was extracted from cells as previously reported [47]. The *hTERT* promoter (region I: from the position −650 to +150 bp relative to the transcription start site (TSS)) and a distal regulatory upstream region reported as an enhancer (region II: from the position −5500 to −4900 bp relative to the TSS) were amplified using Pfu DNA polymerase (Promega) and specific primers whose sequences and localizations are reported in Appendix A. The PCR products were purified using a commercial kit (Nucleospin Gel and PCR Clean Up, Macherey-Nagel SAS, 67722 Hoerdt, France) and sequenced by the Sanger sequencing method (Eurofins, MWG Biotech, Ebersberg, Germany). The DNA sequences of each amplicon are presented in Appendix A.

### 2.4. Bisulfite Modification

Bisulfite methylation analysis was performed basically as previously described [25]. Briefly, bisulfite conversion was performed on 1 µg of purified DNA using the EZ DNA Methylation Kit (ZD5001, Zymo Research, Ozyme, SAS, Saint-Cyr-L’Ecole, France). Bisulfite-converted DNA was used for PCR amplification of the regions of interest with the primers listed in Appendix A. We designed the assay to explore both regions I and II as defined above. Bisearch (http://bisearch.enzim.hu, last updated on 1 December 2020) [48] and Methprimer (http://www.urogene.org/methprimer/, last accessed on 15 June 2017) [49] programs were used to design primers.

To determine the methylation status of individual CpG sites, the PCR amplicons were purified with NucleoSpin Gel and PCR Clean-up Kit (Macherey-Nagel SAS, Hoerdt, France) and subcloned into the pGEM-T Easy vector (A137A, Promega, Charbonnières-les-Bains, France) as described in the manufacturer’s instructions. Competent E. coli (JM109 competent bacteria, Promega, Charbonnières-les-Bains, France) were transformed using the ligation product. Colonies were grown overnight on LB (Luria-Bertani)-Agar containing 32 μg/mL X-gal, 120 μg/mL IPTG, and 100 μg/mL Ampicillin. After white colony selection and checking the DNA insertion by PCR, colonies were incubated overnight for enrichment in LB medium with 100 μg/mL Ampicillin at 37 °C under agitation. Plasmid DNA was isolated using a Nucleospin Plasmid Kit (Macherey-Nagel SAS, Hoerdt, France) and sequenced by Sanger sequencing.

For each experiment, 10–20 plasmid subclones were sequenced (Eurofins, MWG Biotech, Ebersberg, Germany) for the assessment of CpG methylation. DNA sequences were analyzed using ChromasPro software (Technelysium, Australia). Methylation sites were visualized, and quality control was performed using the web-based tool “QUMA” (Riken, Japan, http://quma.cdb.riken.jp, last updated on 15 May 2019) [50].

### 2.5. Statistical Analysis

Results are expressed as mean ± SE. All statistical analyses were carried out using GraphPad Prism (GraphPad Software). All experiments were performed in triplicate unless otherwise specified. The threshold for significance was defined as *p* < 0.05, *p* < 0.01, *p* < 0.001, and *p* < 0.0001 indicated by the symbols (*), by (**), (***), and (****), respectively. Correlations between DNA methylation and *hTERT* gene expression variations were tested using the Spearman correlation coefficient.

## 3. Results

### 3.1. TERT mRNA Expression and Mutational Analysis in Breast Cancer Cell Lines 

In this study, we utilized a panel of breast cancer cell lines representing major breast cancer cell subtypes (Table 1). MCF7, T47D, and ZR75.1 are estrogen receptor-positive (ER^+^); BT474, SKBR3, and MBA-MB-361 are human epidermal receptor 2 amplified and overexpressed (HER2^+^); and MDA-MB-453, MDA-MB-231, SUM159PT, and SUM185PE are triple-negative breast cancer cell lines (TNBC). 

*hTERT* mRNA baseline expression levels were first evaluated. SKBR3 and SUM185PE cells exhibited the highest constitutive *hTERT* expression, whereas BT474 and MDA-MB-361 cells showed the lowest *hTERT* expression (Figure 1).

By direct Sanger sequencing, we then assessed the presence of either *TERT* mutations or single nucleotide polymorphisms (SNPs) in both the *hTERT* promoter (referred to as region I) and a region (referred to as region II) located approximately −5 kb upstream of the *hTERT* transcriptional site start (TSS) and previously described as an enhancer element [51]. Neither recurrent *hTERT* promoter non-coding mutation C228T nor C250T was detected, except for the MDA-MB-231 and SUM159PT cell lines, in which the C228T mutation was identified in its homozygous form (Appendix A). In addition, sequence analysis revealed the presence of already known SNPs (Appendix A). However, no association between the level of baseline *hTERT* expression, the presence of a specific *hTERT* SNP/mutation, or the status of hormonal receptors was observed.

### 3.2. ATRA Induces Differential Modulations of hTERT Expression in Breast Cancer Cell Lines

Next, we investigated the expression of *hTERT* after ATRA treatment. As shown in Figure 2, ATRA treatment induced a significant decrease in *hTERT* expression in ER^+^ cells (with more than 75% repression in MCF7 and T47D cells). In contrast, TNBC cells seem globally resistant to ATRA-induced *hTERT* repression. Notably, in HER2^+^ cells, the effect of ATRA treatment was heterogeneous with a significant *hTERT* repression in SKBR3, and MDA-MB-361 cells, whereas BT474 cells demonstrated a substantial increase in *hTERT* expression.

### 3.3. Pattern of DNA Methylation at hTERT Promoter Region I and Region II

We used bisulfite sequencing to access the methylation profiles of *hTERT* at a CpG-specific resolution in the ten breast cancer cell lines. We focused the analysis on two different regulatory regions of the *hTERT* gene previously analyzed in APL cells [30]: the first region (region I) of the *hTERT* promoter extended from −650 to + 150 bp relative to the transcription start site (TSS), the second region (region II), the *hTERT*-5 kb region, located far upstream from the TSS (−5500 bp to −4900 bp) identified as a putative enhancer domain [51]. 

The CpG methylation profiles are summarized graphically for both *hTERT* regions in Figure 3a,b. As previously described, DNA methylation analysis of region I led us to confirm the existence of two functional domains differentially methylated (Figure 3a), a proximal one close to the TSS that was largely hypomethylated in all the cell lines analyzed, and a distal region further upstream (−200 bp upstream of the TSS), commonly hypermethylated. Among the breast cancer cell lines analyzed, SKBR3 and SUM185PE showed the lowest methylation level of this region. Note that this region covers mostly the previously described 433 bp THOR region [26]. 

Region II, located far upstream from the TSS (−5500 bp to −4900 bp), displayed global hypermethylation except for the SUM159PT cell line (Figure 3b). Upon closer examination, it appeared that the differences in the global level of methylation clustered with the cell molecular classification. Indeed, TNBC cell lines clustered into the less global methylation score for this region compared to the ER^+^ and Her2^+^ cell lines.

### 3.4. Relationship between DNA Methylation and Expression of hTERT at Baseline or after ATRA Treatment

Using Spearman’s rank correlation coefficient, we then assessed the relationship between the baseline expression of *hTERT* and either the global or the individual CpG site methylation levels in both regions I and II (Figure 4). The analysis was carried out, first, considering all cell lines and, second, excluding the HER2^+^ group of cell lines because of their genetic peculiarity, as mentioned earlier.

When all the cell lines were considered, we identified no significant relationship between the constitutive expression of *hTERT* and the DNA methylation levels measured either globally for each region (“average”) or at individual CpG sites. However, when the Her2^+^ group was excluded from the analysis, a significant negative correlation was observed between the sustained expression of *hTERT* and the methylation level of several CpGs located in the distal part of the promoter region I. It can be pointed out that this correlation was not observed if the average levels of DNA methylation of both regions were considered (ρ = 0.46, *p* = 0.30, and ρ= −0.18, *p* = 0.71 for region II and *hTERT* promoter region I, respectively).

We next evaluated whether methylation levels were correlated to the sensitivity to ATRA-induced *hTERT* repression (Figure 5). When all the breast cancer cell lines were considered, the Spearman correlation coefficients obtained indicate that *hTERT* expression after ATRA treatment of cells was not significantly correlated with the average percentage of DNA methylation of both the studied region of the *hTERT* gene (ρ = −0.8, *p* = 0.28, and ρ = −0.027, *p* = 0.44 for region II and *hTERT* promoter region I, respectively). Similarly, no significant correlation was observed in *hTERT* promoter region I when the Her2^+^ group was excluded from the analysis (ρ = −0.18, *p* = 0.71). However, in the case of region II, the exclusion of the HER2^+^ group of cell lines revealed a strong negative correlation between *hTERT* expression after ATRA treatment and the average level of DNA methylation (ρ = −0.964, *p* = 0.0028).

It is important to note that this significant negative correlation also involves several individual CpG sites. Among them, the highest statistically significant correlation was that of a specific CpG (CpG^8^ of region II), located at ch5: 1,300,438 (ρ = −0.991, *p* < 0.0001). Figure 6a,b illustrates the inverse relationship between the methylation level of this specific CpG site and *hTERT* expression after the treatment of cells with ATRA. Note that ATRA-induced repression of *hTERT* was associated with a methylation level of CpG^8^ over 40%, whereas, under this threshold, resistance to this repression was observed.

Altogether, the results indicate that the DNA methylation level of the −5 kb region of *hTERT* and in particular, that of the CpG located at chr5: 1,300,438 could be predictive of the sensitivity of cells to ATRA-induced *hTERT* repression.

## 4. Discussion

Based on the clinical efficiency of ATRA in APL, efforts were made to extend the therapy spectrum of this drug as a single agent or in combination with other therapies in the treatment of some solid tumors including breast cancers. However, due to the complicated and heterogeneous physiopathology of these cancers, ATRA-based therapy strategies might only be effective in some patients. Moreover, the antitelomerase properties of ATRA demonstrated in myeloid leukemia [21,23] were not taken into account in the studies performed in breast cancers. In these conditions, studying the factors associated with ATRA-induced *hTERT* repression might be beneficial to identify subpopulations of patients with breast cancer who could potentially respond to ATRA, in terms of *hTERT* repression. On this basis, it will be possible to propose innovative therapeutic strategies combining ATRA treatment with existing therapies. 

In this study, we used different breast cancer cell lines with distinct molecular characteristics and analyzed the effect of ATRA on *hTERT* expression. These cell lines express different levels of *hTERT* mRNA. We observed distinct responses to ATRA treatment depending on the cell line: either high repression (MCF7, T47D, ZR75.1, SKBR3, MDA-MB-361), a significant increase (MDA-MB-231, SUM159PT, and BT474), or light or even no effect (SUM185PE, MDA-MB-453). Interestingly, the triple-negative breast cancer cell lines are the most resistant to ATRA-induced *TERT* repression. These observations are in agreement with those already published [43,44]. It is important to note that the variability of this response may then explain the failure of the clinical trials conducted using ATRA [41,42]. This supports the idea that being able to accurately predict the response to ATRA-induced *TERT* repression may help in optimizing drug development strategies.

These differences to ATRA treatment observed above were not explained by *hTERT* promoter mutations or SNP, suggesting that other mechanisms responsible for telomerase activation might be involved. Epigenetic alterations are well recognized as a common hallmark of human cancer including breast cancers [52,53].

Initial studies performed to evaluate the role of DNA methylation in the epigenetic regulation of *hTERT* expression have led to contradictory results. One of the reasons is that the majority of these studies were focused on restricted and often distinct regions of the *hTERT* promoter. Our group and others have recently demonstrated that the location of DNA methylation within the promoter is important to consider, DNA methylation not being uniform throughout the promoter. Indeed, in APL cells, we identified a distal domain of *hTERT* promoter whose methylation could modulate the binding of *hTERT* repressors (specifically WT1) and account either for *hTERT* reactivation or for resistance to ATRA-induced *hTERT* downregulation [25]. Later, this domain was referred to as THOR (*TERT* hypermethylated oncological region), involved in the cancer-associated transcription of *TERT* [54,55]. More recently, our studies in APL cells identified, far upstream of the *hTERT* promoter (at about −5 kb from the TSS), a region possibly involved in *hTERT* regulation in cooperation with the *hTERT* promoter [30]. 

We, therefore, focused on DNA methylation profiles as one type of molecular characteristic that may modulate the response to ATRA treatment in breast cancer cells. Methylation was analyzed either globally or on the CpG dinucleotide resolution level. No correlation was found between the constitutive expression of *hTERT* measured in the ten cell lines of this study and the mean of the DNA methylation levels measured either globally for both regions of *TERT* analyzed or at each CpG. Similarly, we found no significant association between changes in *hTERT* expression induced by ATRA treatment and DNA methylation levels measured globally for the two *TERT* regions analyzed.

Importantly, after exclusion of the Her2^+^ cell lines, we found a significant high negative correlation between ATRA-induced *hTERT* regulation and DNA methylation measured at the distal −5 kb region II of the *hTERT* gene. Thus, higher levels of DNA methylation of this region (either on average or at several individual CpG sites) were associated with higher repression of *hTERT* after ATRA treatment. The unique properties of HER2^+^ tumors can explain this observation. Indeed, in these tumors, the constitutive activation of HER2 receptors triggers downstream signaling through multiple pathways including PI3K/AKT/mTOR [33,34,35]. Therefore, in this specific group of breast cancer cells, the importance of *hTERT* DNA methylation as an epigenetic biomarker in response to ATRA treatment may be masked by the presence of these constitutively activated signaling pathways that act as drivers of pathogenesis. This implies that the predictive potential of the methylation status of this region can be type-specific for breast cancer. 

Most of the published works on *hTERT* methylation profiles focused on the promoter region of this gene, generally excluding the involvement of remote regions. One strength of this study was to confirm, in another cellular model, our previous observations reported in the APL cellular model [30] and further demonstrate the importance of the distal −5 kb region of *hTERT* in ATRA-induced *hTERT* repression. However, the exact mechanism in which the methylation status of this region can control the sensitivity to ATRA-induced *hTERT* repression is still unclear and requires further investigation.

While it is clear that the results presented here using cell lines may have limited clinical relevance, they provide the basis for further work to validate them in preclinical models as patient-derived xenografts and translate them to the clinic. Indeed, the implications for therapy are clear in that monitoring the DNA methylation status of the *hTERT* −5 kb region or several individual CpG sites may be useful as predictive biomarkers for the selection of patients most likely to benefit from ATRA treatment. The best predictive goodness of fit was observed for CpG^8^ located in the −5 kb *hTERT* region at chr5: 1,300,438. Thus, tumors with a high level of methylation of this CpG will appear to be more sensitive to ATRA-induced *hTERT* repression. As *hTERT* repression has been demonstrated to promote drug-induced cell death and rescue cell resistance, checking the methylation status of this particular CpG can help in decision-making to include ATRA in combination therapy and thus contributes to a better clinical outcome.

## Figures and Tables

**Figure 1 biomedicines-10-00695-f001:**
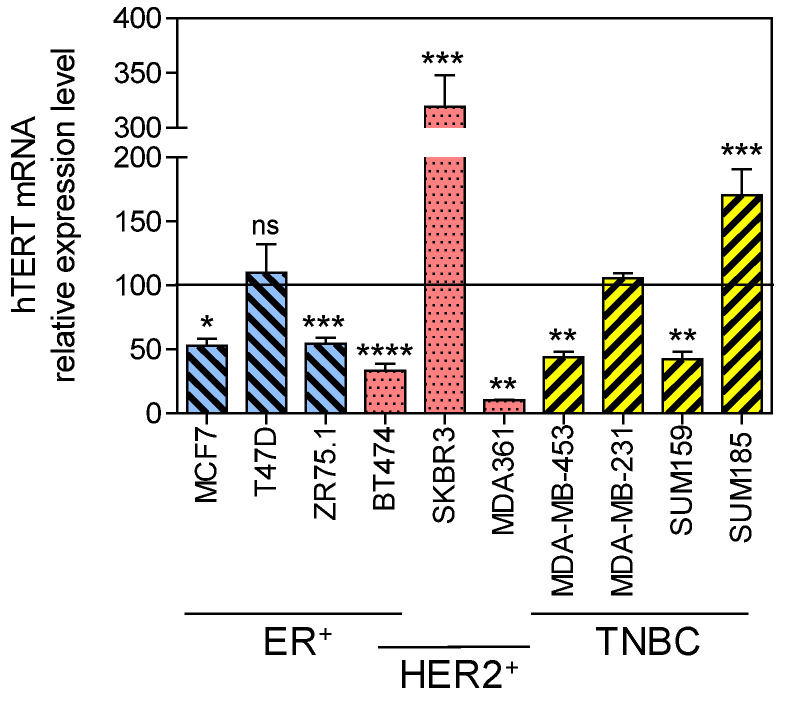
Constitutive *hTERT* gene expression in breast cancer cell lines. *hTERT* mRNA expression levels were determined by quantitative RT-PCR. Expression levels were normalized to the expression levels of *PBGD* and *GAPDH* housekeeping genes and to *hTERT* mRNA level of MDA-MB-231 cells set to 100. One-way ANOVA (*p* = 0.05) Dunnett’s multiple comparison test. ER^+^: estrogen receptor positive, HER2^+^: human epidermal receptor 2 amplified and overexpressed, TNBC: triple-negative breast cancers. * *p* < 0.05; ** *p* < 0.01; *** *p* < 0.001; **** *p* < 0.0001; ns, non-significant.

**Figure 2 biomedicines-10-00695-f002:**
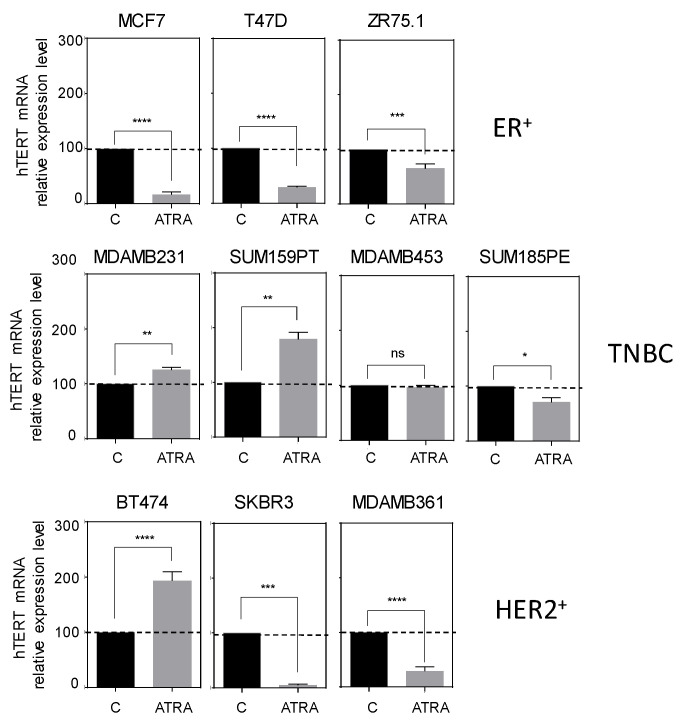
*hTERT* mRNA expression upon ATRA treatment of breast cancer cell lines. Histograms represent the relative expression of *hTERT* in ATRA-treated cells compared to non-treated cells. The normalized *hTERT* expression of non-treated cells was set to 100. t-Student unpaired two-tailed. * *p* < 0.05; ** *p* < 0.01;*** *p* < 0.001; **** *p* < 0.0001; ns, non-significant.

**Figure 3 biomedicines-10-00695-f003:**
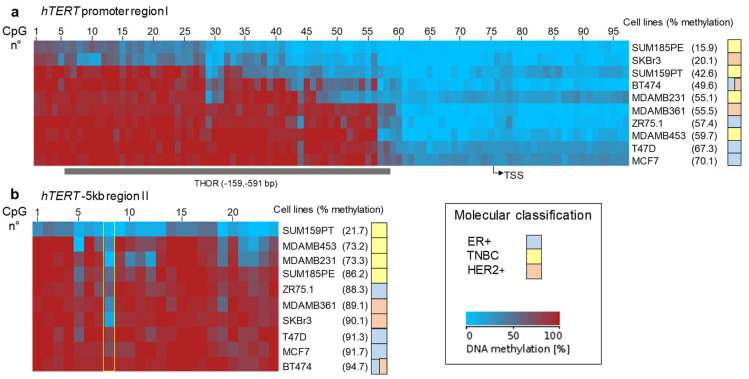
Condensed graphical representation of *TERT* methylation patterns in breast cancer cell lines. (**a**) *hTERT* promoter (region I). (**b**) *hTERT* −5 kb region upstream the TSS (region II). The methylation patterns were created using BDPC compilation software: columns represent CpG sites, while rows represent the average methylation profile of each PCR product for each cell line. The average level of CpG methylation is indicated in brackets. In (**a**)**,** the position of THOR between −159 and −591 bp from the TSS is shown. In (**b**), the position of the remarkable CpG (CpG^8^, chr5: 1,300,438) is pointed out. THOR: *TERT* hypermethylated oncologic region. TSS: transcription start site.

**Figure 4 biomedicines-10-00695-f004:**
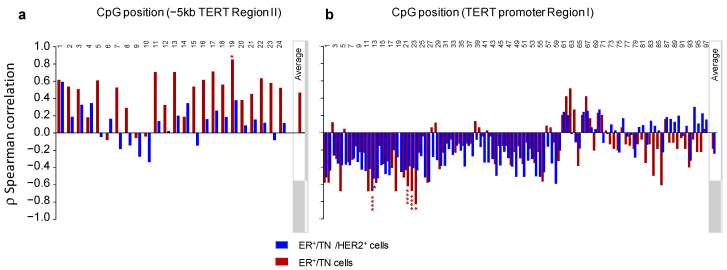
*ρ* Spearman correlations between baseline *TERT* expression and CpG methylation levels for either *TERT* region II (**a**) or *TERT* promoter region I (**b**). Blue bars: analysis performed considering all cell lines. Red bars: HER2^+^ cells were excluded from the analysis. CpG positions of each studied region are indicated. “Average”: the average DNA methylation level for each region is specified in the grey box. Corresponding *p*-values are shown with each bar: * *p* < 0.05; **** *p* < 0.0001.

**Figure 5 biomedicines-10-00695-f005:**
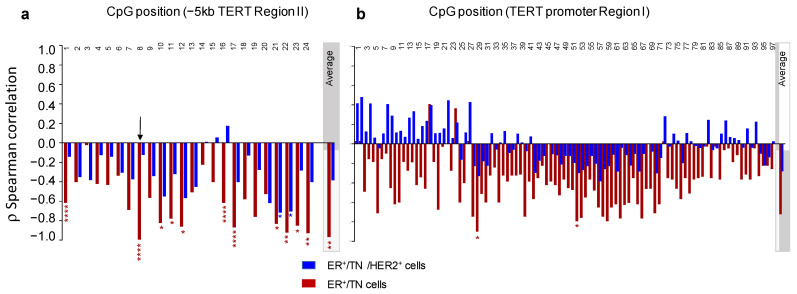
*ρ* Spearman correlation between *TERT* regulation after ATRA treatment of cells and CpG methylation levels for either −5 kb *TERT* region II (**a**) or *TERT* promoter region I (**b**). Blue bars: analysis was performed considering all cell lines. Red bars: HER2^+^ cells were excluded from the analysis. CpG positions of each studied region are indicated. In (a), the position of the remarkable CpG^8^ (chr5: 1,300,438) is pointed out. “Average”: the overall methylation status for each region is specified in the grey box. Corresponding *p*-values are shown with each bar: * *p* < 0.05; ** *p* < 0.01; **** *p* < 0.0001.

**Figure 6 biomedicines-10-00695-f006:**
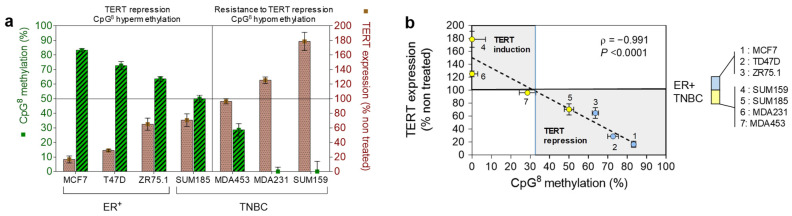
Histograms (**a**), scatter plots, and linear regression line (**b**) for *hTERT* expression induced by ATRA treatment vs. methylation of the CpG located in region II at chr5: 1,300,438 (CpG^8^). The dashed line indicates the fitted line from the linear regression model. The vertical blue line indicates the threshold methylation level above which ATRA-induced *TERT* repression may be expected. Blue and yellow dots represent the ER^+^ and TNBC cell lines, respectively.

**Table 1 biomedicines-10-00695-t001:** Breast cancer cell lines used in the study. ^1^ Conflicting reports with ATCC indicating TNBC and some publications indicating HER2 amplified, although not highly expressed [46]. ^2^ Molecular classification: ERα^+^ (estrogen receptor alpha expressing), HER2^+^ (HER2 amplified), TNBC (triple-negative, lacking ER, PR, and HER2).

Cell Line	MCF7	T47D	ZR75.1	BT474	SKBR3	MDAMB361	MDAMB453 ^1^	MDAMB231	SUM159PT	SUM185PE
Tumor type	Adeno carcinoma	Ductal carcinoma	Ductal carcinoma	Ductal carcinoma	Adeno carcinoma	Adeno carcinoma	Metaplastic carcinoma	Adeno carcinoma	Anaplastic carcinoma	Ductal carcinoma
	Epithelial	Epithelial	Epithelial	Epithelial	Epithelial	Epithelial	Epithelial	Mesenchymal	Mesenchymal	Epithelial
Molecular ^2^ classification	ERα^+^			ERα^+^/ HER2^+^	HER2^+^	HER2^+^	TNBC			
*hTERT* promoter status	Wild-type	Wild-type SNP T349C	Wild-type SNP T349C	Wild-type SNP T349C	Wild-type	Wild-type	Wild-type	Mutated C228T	Mutated C228T SNP T349C	Wild-type SNP T349C

## Data Availability

Not applicable.

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
