# Peer review of "hTERT DNA Methylation Analysis Identifies a Biomarker for Retinoic Acid-Induced hTERT Repression in Breast Cancer Cell Lines"

_biomedicines, 2022, doi:10.3390/biomedicines10030695_

Round 1

Reviewer 1 Report

The paper titled “hTERT DNA methylation analysis identifies a biomarker for retinoic acid-induced hTERT repression in breast cancer cell lines” is an interesting work that perfectly investigates the effects of all-trans retinoic acid (ATRA) in a subset of breast cancer cell lines and assesses the mutational status of hTERT promoter and the methylation patterns at single CpG resolutioin. I strongly recommend the publication of this paper.

Reviewer 2 Report

The manuscript is well written and presented in a logical way. However, these should be corrected for further proceedings. 

Please separate the aims and objectives of the study from the introduction part.

2. Materials and Methods 

2.1. Cell culture and ATRA treatment: What were the name of the medium and the amount of serum in the culture medium? Instead, it is written as, "harvested using 0.25% trypsin-EDTA", is this information correct? I would suggest revising and correcting. 

Figures 1 and 6: Please use patterned grids for the graphs, so that they can be distinguishable in black and white printing. 

At the end of the discussion, it would be ideal to add the strengths and limitations of the current study. 
